# The Effect of *Sargassum fusiforme* and *Fucus vesiculosus* on Continuous Glucose Levels in Overweight Patients with Type 2 Diabetes Mellitus: A Feasibility Randomized, Double-Blind, Placebo-Controlled Trial

**DOI:** 10.3390/nu16121837

**Published:** 2024-06-12

**Authors:** Karlijn A. M. Geurts, Sjoerd Meijer, Jeanine E. Roeters van Lennep, Xi Wang, Behiye Özcan, Gardi Voortman, Hongbing Liu, Manuel Castro Cabezas, Kirsten A. Berk, Monique T. Mulder

**Affiliations:** 1Department of Internal Medicine, Erasmus University Medical Center, Doctor Molewaterplein 40, 3015 GD Rotterdam, The Netherlands; k.geurts@erasmusmc.nl (K.A.M.G.);; 2Key Laboratory of Marine Drugs, Chinese Ministry of Education, School of Medicine and Pharmacy, Ocean University of China, Qingdao 266100, China; 3Department of Internal Medicine, Franciscus Gasthuis & Vlietland, Schiedamse Vest 180, 3011 BH Rotterdam, The Netherlands

**Keywords:** seaweed, type 2 diabetes mellitus, body composition, cardiovascular risk factors, *Sargassum fusiforme*, *Fucus vesiculosus*

## Abstract

Background: Brown seaweed is promising for the treatment of type 2 diabetes mellitus (T2DM). Its bioactive constituents can positively affect plasma glucose homeostasis in healthy humans. We investigated the effect of the brown seaweeds *Sargassum (S.) fusiforme* and *Fucus (F.) vesiculosus* in their natural form on glucose regulation in patients with T2DM. Methods: We conducted a randomized, double-blind, placebo-controlled pilot trial. Thirty-six participants with T2DM received, on a daily basis, either 5 g of dried *S. fusiforme*, 5 g of dried *F. vesiculosus*, or 0.5 g of dried *Porphyra* (control) for 5 weeks, alongside regular treatment. The primary outcome was the between-group difference in the change in weekly average blood glucose levels (continuous glucose monitoring). The secondary outcomes were the changes in anthropometrics, plasma lipid levels, and dietary intake. The data were analyzed using a linear mixed-effects model. Results: The change in weekly average glucose levels was 8.2 ± 2.1 to 9.0 ± 0.7 mmol/L (*p* = 0.2) in the *S. fusiforme* group (n = 12) and 10.1 ± 3.3 to 9.2 ± 0.7 mmol/L (*p* = 0.9) in the *F. vesiculosus* group (n = 10). The between-group difference was non-significant. Similarly, no between-group differences were observed for the changes in the secondary outcomes. Discussion: A daily intake of 5 g of fresh, dried *S. fusiforme* or *F. vesiculosus* alongside regular treatment had no differential effect on weekly average blood glucose levels in T2DM.

## 1. Introduction

Over the last decades, diabetes mellitus has become a worldwide pandemic. Of approximately 537 million diabetes cases worldwide, 90% represent type 2 diabetes mellitus (T2DM). The development of T2DM is often caused by chronic positive energy balance, and its increased prevalence is accompanied by the increasing prevalence of obesity. Nutritional interventions may prevent or even reverse T2DM [1,2,3,4,5].

Specific species of brown seaweed might be an important part of such a nutritional intervention. Examples of brown seaweed species are *Sargassum fusiforme* (*S. fusiforme*, also called ‘Hijiki’), which grows along the coastlines of East Asia, and *Fucus vesiculosus* (*F. vesiculosus*), which is a seaweed found on the coasts of the North Sea, the western Baltic Sea and the Atlantic and Pacific Oceans. Brown seaweeds contain a high amount of health-promoting compounds, including soluble and insoluble dietary fibers, polysaccharides, ω3-polyunsaturated fatty acids, carotenoids, polyphenols, fat-soluble vitamins and minerals, and phytosterols, like fucosterol, saringosterol, and the carotenoid fucoxanthin [6,7,8,9,10]. The bioactive constituents of brown seaweeds were found to improve glucose tolerance, regulate blood lipids, and enhance feelings of satiety and, thereby, contribute to the prevention of weight gain, which are all important in the treatment of people with T2DM [11,12,13,14,15,16,17].

In animal models, positive effects were observed for *S. fusiforme* and for a combination of *Ascophyllum nodosum* and *F. vesiculosus*. *S. fusiforme* administration (100 mg/kg, 4 weeks) alleviated hyperglycemia in diabetic mice [18,19]. The administration of an extract of *Ascophyllum nodosum* and *F. vesiculosus* reduced both overall and postprandial blood glucose levels in a mouse model of non-alcoholic steatohepatitis [12]. The mechanisms of action underlying these effects of seaweed components on glucose regulation may be multiple. It may decrease the absorption of glucose as a result of inhibitory effects on the major intestinal carbohydrate-hydrolyzing enzymes α-amylase and α-glucosidase [12,20,21,22]. Seaweed extracts were reported to inhibit α-glucosidase and α-amylase activity but did not affect fasting blood glucose in obese individuals [23]. Increased fiber intake from seaweed may also improve glucose regulation via beneficially modulating the gut microbiome, resulting in an increased production of short-chain fatty acids [24,25,26]. In addition, evidence was obtained for the regulation of lipid metabolism and satiety hormones in animal models [14]. Seaweed-derived fucoidan reduces plasma triglycerides, total cholesterol, and LDL levels in high fat-fed mice. Also, fucoidan inhibits adipogenesis and reduces leptin levels in obese mice with hyperleptinemia. Moreover, evidence was obtained that fucoxanthin supplementation can reduce body weight and white adipose tissue weight in mice [14].

Up to now, clinical studies addressing the beneficial effects of daily consumption of seaweed in T2DM patients are limited. A recently published meta-analysis on the effect of brown seaweed consumption for blood glucose management showed that postprandial blood glucose, glycated hemoglobin (HbA1c), and Homeostatic Model Assessment of Insulin Resistance (HOMA-IR) levels significantly improved in the seaweed group compared to the control group [27]. Although promising, the majority of studies used supplements with high dosages of seaweed extracts. To find out whether the use of seaweed in its natural form as part of a healthy diet can also contribute to improving glucose regulation, pragmatic dietary trials are needed. Therefore, the aim of the present study was to determine the effect of daily consumption of feasible amounts of the brown seaweed *S. fusiforme* or *F. vesiculosus* on top of the habitual diet and medication on average blood glucose levels in overweight adults with T2DM. In addition, we examined the effect of the administration of *S. fusiforme* or *F. vesiculosus* on bodyweight, self-reported energy intake, and risk factors for cardiovascular disease (CVD).

## 2. Materials and Methods

A pilot randomized, placebo-controlled, double-blind, single-center trial was conducted at the Franciscus Gasthuis & Vlietland hospital in Rotterdam, The Netherlands. The eligible participants were adult patients (>18 years old) with T2DM, based on the criteria of the American Diabetes Association (ADA), and a BMI > 25 kg/m^2^. The exclusion criteria were type 1 or monogenetic forms of diabetes, thyroid disease, pregnancy, usage of corticosteroids, usage of blood anti-coagulants, history of heart failure or myocardial infarction within the last 3 months, transplantation, or an allergy to shellfish.

A total of 37 patients from the diabetes outpatient clinic of the Franciscus Gasthuis & Vlietland in Rotterdam signed an informed consent form before study enrollment.

The protocol (NL66189.078.18) was approved by an independent ethics committee: Medisch-ethische toetsingscommissie (METC) of the Erasmus MC, Rotterdam, The Netherlands. In addition, approval was received from the institutional board of Erasmus MC and Franciscus Gasthuis & Vlietland. This study was conducted in accordance with the Declaration of Helsinki and its revisions and the International Conference on Harmonization (ICH) guidelines for Good Clinical Practice (GCP), governing the conduct of studies, and all applicable local regulations.

The intervention was a daily intake of 5 g of dried *S. fusiforme* or 5 g of dried *F. vesiculosus*, while the control/placebo group received 0.5 g of dried *Porphyra yezoensis* (*Porphyra*, also called ‘nori’). Dried crude *S. fusiforme* was harvested in a fixed area in Zhejiang Province in China in May 2018 and provided by Zhejiang Yukang Food Co., Ltd., Laiwu, China. Harmful unwanted heavy metals were removed from the brown seaweed according to the protocol of Yamashita et al. [28] and, after drying for 16 h at 55 °C, the brown seaweed was stored in a cool place out of direct sunlight until usage. Although the composition with respect to bioactive compounds may differ between fresh and dried seaweed, many bioactive components with health-improving properties have been demonstrated to remain intact. Among the compounds of interest, the LXR-activating sterols, remain intact after drying [29,30]. *F. vesiculosus* was provided by former Seaweed Harvest Holland/St. Zeeschelp, Barendrecht, The Netherlands, and collected in the Eastern Scheldt, a tidal bay connected with the North Sea in The Netherlands. The tips of the *S. fusiforme* and *F. vesiculosus* were collected and pre-treated (washed and dried). The detailed quality control for both *S. fusiforme* and *F. vesiculosus* was performed by Prof. Liu, School of Medicine and Pharmacy, Laboratory for Marine Drugs and Bioproducts, Ocean University of China, Qingdao National Laboratory for Marine Science and Technology, Qingdao, China. In this manner, a quality stable sample could be used to perform the experiments.

A daily intake of 0.5 g of dried *Porphyra*, purchased from Terrasana BV, Leimuiden, The Netherlands, was used in the control group. *Porphyra* was used as the control/placebo because of its similar appearance and taste. However, *Porphyra* is not a brown but rather a red algae and contains, per gram, only 1/10 of the dosage of the bioactive compounds found in *S. fusiforme* and *F. vesiculosus*. In Table 1, a summary of the composition of the seaweeds given to the participants in this study can be found.

The participants were given randomization numbers in chronological order, based on the date of their first visit, via a computer-generated randomization list. The participants were randomly assigned (1:1:1) to receive either *Porphyra* (control), *S. fusiforme*, or *F. vesiculosus*. The study participants, site personnel, sponsors, and clinical research staff were blinded for the treatment assignment. Visually, no distinction could be made between the treatments, as the storage box and the bags inside the box containing the seaweed were identical in appearance.

An overview of the study procedures is depicted in Figure 1. After inclusion, the participants continued their habitual diet for one week, after which baseline measurements were obtained. The participants visited the outpatient clinic weekly for a total of seven times over 6 weeks in the morning before they had their breakfast. Blood samples, weight/height measurements, systolic and diastolic blood pressure, and medical history were collected at the first (week 0) and last visit (week 5). Blood glucose was monitored continuously during this 6-week period. Moreover, the participants were asked to report their weekly nutritional intake at week 0, week 2, and week 5 of the trial in a nutritional diary. From week 1 onward, the study participants were instructed to consume 5 g of *S. fusiforme*, 5 g of *F. vesiculosus*, or the control (0.5 g *Porphyra*) daily for 5 weeks. Prior to consumption, the seaweeds needed to be soaked in boiling water for 2–3 min. The timeframe of intake was fixed between 5 pm and 8 pm. The intake of seaweeds was according to the preference of the participants, either mixed with a meal or separate. Recipes were provided to improve compliance.

Blood glucose was monitored continuously. The participants received an iPRO^TM^2 (Medtronic^®^, Dublin, Ireland) continuous glucose monitor (CGM) at the first study visit. The participants also had to perform a manual blood glucose test 3 times a day during their entire study participation for calibration purposes. The iPRO^TM^2 was renewed weekly during the visits by the research nurse, and the results were blinded for the participants. From the CGM data, the time in range (TIR), time above range (TAR), time under range (TUR), and 2 h postprandial glucose levels were determined.

The primary outcome was the between-group difference in weekly average glucose levels measured during week 0, when the usual diet was consumed, and weeks 1 to 5 during daily seaweed consumption. Plasma glucose was monitored every 5 min as described previously by an iPRO^TM^2 CGM (Medtronic^®^), which was blinded for the participants. The secondary outcomes were the estimated HbA1c (eHbA1c; estimated from the CGM measurements), TIR, TAR, TUR, and 2 h postprandial glucose levels that were derived from the glucose data by using Carelink^TM^ system (Medtronic^®^, version 2020). The secondary outcome parameters were anthropometrics, cardiovascular risk factors, dietary intake, and intake of antidiabetic drugs. The anthropometric parameters were waist circumference (cm), bodyweight (kg), and body mass index (BMI; kg/m^2^). Cardiovascular risk factors were determined using standard laboratory measurements for plasma total cholesterol (TC [mmol/L]), triglycerides (TG [mmol/L]), HDL cholesterol (HDLc [mmol/L]), LDL cholesterol (LDLc [mmol/L]), lipoprotein (a) (Lp(a) [g/L]), apoB100 (g/L), and blood pressure (mmHg). Dietary intake was recorded in food diaries at baseline (week 0), week 2, and week 5. The nutritional intake was quantified using ‘Eetmeter’ (Voedingscentrum, version 4.7). Antidiabetic medication prescriptions before the intervention were compared to the prescriptions post-intervention and scored by a consultant as either unchanged, less medication, or more medication. The most recent prescriptions of the participant were used before entering the study, and the prescriptions during the study until one month after the intervention were used to evaluate changes in the participant’s treatment. Compliance was monitored by analyzing the marine phytosterols, fucosterol, and saringosterol, contained by *S. fusiforme* and *F. vesiculosus*. In addition, plant sterols and cholesterol, its precursors, and metabolites were measured. Sterol concentrations were determined using GC-MSMS [31]. Plant sterols, fucosterol, saringosterol, campesterol (mg/dL), stigmasterol (µg/dL), sitosterol (mg/dL), avenasterol (µg/dL), brassicasterol (µg/dL), cholesterol (GC [mg/dL]) and its precursors, lathosterol (mg/dL), lanosterol (mg/dL) and desmosterol (mg/dL) and metabolites, 24-hydroxycholesterol (ng/mL), 7α-hydroxycholesterol (ng/mL), and 27-hydroxycholesterol (ng/mL) were analyzed in plasma samples at baseline and post-intervention. The campesterol/cholesterol ratio was determined as a proxy for cholesterol absorption in the intestine, and the lathosterol/cholesterol ratio was determined as a proxy for cholesterol synthesis.

The optimal target sample size of 12 participants per treatment group for a pilot study was derived from a report by Julious et al. [32]. Due to the pandemic of COVID-19, only 36 participants were enrolled in this study, leaving no room for study withdrawals and a per-protocol analysis.

For the statistical analyses, SPSS Statistics for Windows, version 24.0 (IBM Corp., Armonk, NY, USA) was used. According to their distribution, the data were either expressed as mean ± SE or median (interquartile range [IQR]). Further, the data were tested for normality with a Shapiro–Wilk test, and heterogeneity was tested by a Levene’s test. Abnormal datasets (residuals) were corrected using log transformation before analyses. Outliers were replaced (corrected) by the median value ± 1.282 × standard deviation (the value 1.282 reflects a z-score fitting an 80% confidence interval).

According to their distribution, correlations were analyzed with a parametric test (Pearson) or a non-parametric test (Spearman). Within-group differences were, according to their distribution, analyzed using a parametric test (one-way ANOVA) or non-parametric test (Kruskal–Wallis test). Mixed-effects models were used to determine between-group differences over time. The model covariance type was set at variance components. The study participants were placed in the model as random effects, and group and time were set as fixed factors. Mixed-effect models were run separately for the comparison between *S. fusiforme* and the control or *F. vesiculosus* and the control. Changes in antidiabetic medicine use were analyzed using a chi-square association test. The TIR, TAR, and TUR were analyzed by a negative binominal Poisson regression. When log transformation was not possible given the zero-inflation, a zero-inflated negative binomial regression was used to analyze the data. The significance level was set at *p* < 0.05.

## 3. Results

Out of a total of 114 participants that were found eligible for inclusion, 37 signed the informed consent form. A flow chart of the participant allocation and study participation is depicted in Figure 2. Among the 37 participants, there was one screen failure due to the inclusion of a participant with a BMI < 25 kg/m^2^. A total of 36 participants were randomized into the three study groups and underwent baseline measurements. Nine participants withdrew from the study at different time points. The reasons for withdrawal were (1) gastrointestinal complaints (n = 3; *F. vesiculosus*), (2) the study being too demanding (n = 4; 2 control, 1 *F. vesiculosus*, 1 *S. fusiforme*), (3) allergic reaction to the band-aid used for the CGM (n = 1; *F. vesiculosus*), and (4) terminal illness of a family member (n = 1; *S. fusiforme*). Twenty-seven participants finished the study. The intention-to-treat analysis included 36 study participants.

The baseline characteristics are shown in Table 2. At baseline, we found that waist circumference differed significantly (*p* < 0.01) between all groups, and triglycerides differed significantly between the intervention groups (*p* < 0.05). Men represented 69% (n = 25) of the study population. The median age was 65 years (IQR 57–74), and the median duration of T2DM was 13 years (IQR 8–18). Off the study, 61% (n = 22) of the study population used insulin on a daily basis. Based on the BMI, 33% of the study population could be classified as overweight (n = 12), and 67% of the study population could be classified as obese (n = 24).

### 3.1. The Effect of S. fusiforme and F. vesiculosus Intake on Blood Glucose

Table 3 shows the effects of the administration of *S. fusiforme*, *F. vesiculosus*, or the control on glucose parameters as determined by continuous glucose measurement. The change in weekly average glucose levels throughout the intervention is shown in Figure 3.

The within-group differences of glucose levels between week 1 and week 5 were 8.2 mmol/L to 9.0 mmol/L in *S. fusiforme* and 10.1 mmol/L to 9.2 mmol/L in *F. vesiculosus*, which was not significant. We found no between-group differences for the interventions with the control group (*p* = 0.25 (*S. fusiforme*) and *p* = 0.90 (*F. vesiculosus*)).

We found no within-group differences of 2 h postprandial glucose levels (*S. fusiforme* (*p* = 0.92) and *F. vesiculosus* (*p* = 0.37)). The between-group differences with the control group were also not significant (*p* = 0.97 and *p* = 0.92, respectively) over time.

All three groups showed a higher median TIR at week 5 (control [+7%], *S. fusiforme* [+2%], and *F. vesiculosus* [+3%]). However, no significant effect was found when looking at the between-group differences over time compared with the control group (*S. fusiforme* (*p* = 0.87) and *F. vesiculosus* (*p* = 0.75)). Higher median TAR values were observed in the *F. vesiculosus* group at both week 2 and week 5 of intervention. However, the change over time was not significantly different from the control group (*p* = 0.86). Moreover, there was no significant difference within the *F. vesiculosus* group looking at eHbA1c (*F. vesiculosus p* = 0.82), but when looking at the between-group difference of *S. fusiforme* and the control over time, there was a near-significant difference (*p* = 0.05).

### 3.2. Effect of Seaweed on Use of Medication, Anthropometrics, and Energy Intake

In the control group, 42% (n = 5) of the participants achieved a reduction in the use of antidiabetic drugs at the end of the study. Moreover, a reduction in the use of antidiabetic drugs was observed in 50% (n = 5) in the *S. fusiforme* group, which was not significantly different compared to the control group (χ^2^ (2) = 1.09, *p* = 0.6). A reduction in the usage of antidiabetic drugs relative to baseline was reported significantly less often in the *F. vesiculosus* group as compared to the control group (χ^2^ (2) = 8.23, *p* < 0.05).

Oral administration of *S. fusiforme* or *F. vesiculosus* for 5 weeks did not affect body weight (*p* = 0.7, *p* = 0.6) nor BMI (*p* = 0.9, *p* = 0.3) as compared to the control group. In the *S. fusiforme* group, the mean body weight was reduced, though not significantly (*p* = 0.5). It must be noted that the between-group difference over time of waist circumference was significantly different between *S. fusiforme* and the control (+3.9 cm [95%CI 1.0, 6.7] vs. −6.4 cm [95%CI −0.34, −12.4], *p* = 0.003).

In the intervention groups, there was a non-significant, self-reported decrease in energy intake in comparison with the control group (1748 ± 318 to 1603 ± 292 kcal for *S. fusiforme* and 1428 ± 595 to 1231 ± 484 kcal for *F. vesiculosus*). There were no significant between-group differences when the intervention groups were compared to the control (*S. fusiforme p* = 0.4 and *F. vesiculosus p* = 0.7). No decrease in body weight was observed in both intervention groups.

### 3.3. Effect of S. fusiforme and F. vesiculosus on Plasma Lipids

Plasma lipids at baseline and during treatment are shown in Table 4. After 5 weeks of treatment, a minor reduction in mean TC levels was observed in the *S. fusiforme* group (4.0 mmol/L ± 1.2 to 3.8 mmol/L ± 0.2, *p* = 0.17); in the control group, we also observed a minor but significant reduction (4.1 mmol/L ± 0.8 to 3.9 mmol/L ± 0.3, *p* < 0.05) as compared to baseline (between-group difference, *p* = 0.48). TG levels changed significantly in all three groups but in different directions. The control group had a decrease of 0.3 mmol/L (*p* < 0.05), the *S. fusiforme* group had an increase of 0.3 mmol/L (*p* < 0.001), and the *F. vesiculosus* group showed a minor increase of 0.1 mmol/L (*p* < 0.05).

The *S. fusiforme* group showed higher median HDLc levels post-treatment (1.0 mmol/L [IQR 0.8–1.4] to 1.2 mmol/L [IQR 0.7–1.5], *p* = 0.91), whereas no effects were observed on the median in the control and *F. vesiculosus* groups. The between-group differences were non-significant (*p* = 0.51 and *p* = 0.93, respectively). Looking at plasma Lp(a) concentrations, no significant differences were found within the groups and between the intervention groups compared to the control group. Within the groups, there were some minor but significant changes in ApoA-I levels for all three groups but in different directions: control, *S. fusiforme*, and *F. vesiculosus*, respectively (ApoA-I: 1.5 (1.2–1.8), *p* < 0.001, 1.3 (1.2–1.4), *p* < 0.05 and 1.3 (1.2–1.6), *p* < 0.05). ApoB also displayed minor within-group changes in different directions, but only for the control and *F. vesiculosus*, they were significant (0.8 ± 0.08, *p* < 0.001 and 0.6 ± 0.09, *p* < 0.05), and the between-group differences with the control were non-significant (ApoA-I: *p* = 0.78, *p* = 0.44 and ApoB *p* = 0.57, *p* = 0.89).

### 3.4. Effect of Brown Seaweed on Plasma Sterol Composition

Plasma sterols, including the seaweed-derived fucosterol and saringosterol, were analyzed to monitor compliance with seaweed intake. However, following intake of the seaweed, the plasma fucosterol and saringosterol concentrations were below the detection limit. Intake of *S. fusiforme* was associated with an increase in campesterol (0.48 mg/dL ± 0.06 at baseline to 0.52 mg/dL ± 0.07), whereas a decrease was observed in the control group (0.46 mg/dL ± 0.07 to 0.36 mg/dL ± 0.06, between-group difference *p* = 0.017 (Table 5). Furthermore, brassicasterol levels in the *S. fusiforme* group increased from 8.6 µg/dL (IQR 5.2–12) at baseline to 12 µg/dL (IQR 9.7–14); this was significantly different from the decrease (10 µg/dL [IQR 7.5–13] vs. 6.4 µg/dL [IQR 3.9–8.9]) in the control group (*p* = 0.003). Following 5 weeks of intervention, lower desmosterol levels were observed in the *S. fusiforme* group (0.16 mg/dL [IQR 0.07–0.25] vs. 0.11 mg/dL [IQR 0.07–0.16]) as compared to the baseline levels (*p* < 0.001). However, similar effects were observed in the control group (0.16 mg/dL [IQR 0.06–0.27] vs. 0.12 mg/dL [IQR 0.06–0.17]). No significant changes were observed for desmosterol. Moreover, all groups had decreased levels of 24-OHcholesterol, 7α-OH-cholesterol, 27-OHcholesterol, and cholesterol (GC) following 5 weeks of intervention. However, only between-group differences over time were found between the control and the *S. fusiforme* groups for 27-OHcholesterol and cholesterol (GC) (*p* < 0.001 and *p* < 0.0125, respectively).

## 4. Discussion

In this randomized, double-blind, placebo-controlled pilot study, we found that 5-week intake of 5 g of *S. fusiforme* or *F. vesiculosus* in overweight T2DM patients in addition to their habitual diet and medication did not affect their weekly average blood glucose levels when measured continuously, although eHbA1c was significantly different between the control and *S. fusiforme* groups in favor of the control group. When looking at anthropometrics, only waist circumference was different between the control and the *S. fusiforme* groups in favor of the control group. There were no between-group differences in plasma lipid levels. Plasma triglycerides and ApoA-I as well as 27-hydroxycholesterol changed significantly within the groups upon administration of *S. fusiforme* and *F. vesiculosus* but also in the control group. The usage of antidiabetic drugs decreased in 42% of the participants in the control group and in 50% of the *S. fusiforme* group following 5 weeks of intake, which may have affected the plasma glucose and lipid levels.

The lack of effect of brown seaweed administration on glucose metabolism is unexpected and not in line with the recently published meta-analysis of Kim et al. [27] that showed that supplementation with different brown seaweed species led to significant improvements of postprandial blood glucose, HbA1c, and Homeostatic Model Assessment of Insulin Resistance (HOMA-IR). In this meta-analysis, most studies included were using seaweed extracts, and only three included studies used fresh seaweed [33,34,35]. None of those three studies included *S. fusiforme* or *F. vesiculosus* but utilized the brown seaweed *Undaria pinnatifida*. In subgroup analyses, the use of fresh seaweed was related to improved postprandial blood glucose levels, contrary to our findings. The study population consisted of healthy or overweight people without T2DM, which is different from our T2DM population. Hypothetically, the advanced state of disease might call for a more intensive, higher-dose intervention to be effective. Apart from the use of a different brown seaweed species, the dosage was similar (5 g/day), while the duration of those interventions was even shorter (acute effects or 17-day intervention) then ours. This suggests that the specific brown seaweed used (*Undaria pinnatifida*) was more potent than the seaweeds we used in our study. On the other hand, the use of *F. vesiculosus* in the form of extracts was found to be more effective at improving blood glucose levels than the control treatments. Our participants may not have ingested the recommended 5 g/day because there was evidence of non-compliance in this very group. Three people in the *F. vesiculosus* group stopped the intervention because of gastrointestinal complaints, while in the *S. fusiforme* and control groups, there were no withdrawals related to gastrointestinal complaints. Therefore, it could be that *F. vesiculosus* is not as well-tolerated in its natural form, so that the actual dose taken was much lower and did not result in health effects.

We did observe some positive effects in the control group, which was unexpected and could potentially explain the lack of between-group differences. The modest lower weekly average glucose levels in combination with a reduction in the usage of antidiabetic drugs, as observed in the control group, favors a possible positive effect of the intervention approach itself, such as continuous monitoring of glucose and weekly visits to the hospital, independent of the treatment. However, the *S. fusiforme* group did not show such effects. It cannot be excluded that daily intake of small amounts as low as 0.5 g/day of the red macro algae *Porphyra yezoensis* that we used as a control may have induced the observed beneficial effects. Though beneficial effects of *Porphyra* on blood glucose parameters have been reported, positive effects of only 0.5 g/daily were not anticipated. Several reports indicate that *Porphyra* might be positively related to lower blood glucose levels [36]. The daily intake of at least 8.6 g per day of edible algae, such as the red algae *Porphyra yezoensis* and brown algae *Undaria pinnatifida*, has been associated with a lower incidence of T2DM in Korean men [37]. The red algae contain classes of polysaccharides that can exert strong inhibitory effects on α-amylase in rats [38] and were found to suppress postprandial blood glucose levels in healthy and diabetic rats [36,39,40,41]. Therefore, it cannot be excluded that *Porphyra* exerted a positive effect on blood glucose homeostasis in our study participants; however, it is unlikely due to the low dosage. The intake of the red algae *Porphyra* was possibly better tolerated than the intervention seaweeds in this trial, as the dosage was lower and had less smell.

In line with our finding of reduced circulating cholesterol concentrations upon intake of *Porphyra* and *S. fusiforme*, various human and animal studies have shown the cholesterol-lowering effects of brown algae [42,43,44,45,46,47,48]. Overall, brown seaweeds decrease levels of TC, TG, and LDLc, while those of HDLc increase when used in amounts of 2 g/day dried seaweed [49]. Accordingly, we observed an increase in HDLc upon *S. fusiforme* intake. The phytosterols in brown algae may reduce plasma cholesterol by competing with cholesterol uptake in the intestine in addition to activating the liver X receptor (LXR), resulting in an increased excretion of cholesterol in the feces [50]. Moreover, supplementation of rats fed a high-fat diet with fucoxanthin, prevalent in brown seaweeds such as *S. fusiforme* and *F. vesiculosus*, at 0.2% of dietary intake decreased TG and TC while enhancing their fecal excretion [51,52]. In line with these findings, our data showed a slight reduction in total cholesterol and 27-hydroxycholesterol and a slight increase in HDLc following *S. fusiforme* intake. However, a decrease in total cholesterol and 27-hydroxycholesterol was also observed in the control group. The relatively small effects on plasma lipids may be due to the low daily amount of seaweed consumed or to the fact that the consumption of seaweed was on top of the habitual diet and medication of the participants. The absence of effects on plasma lipids in the *F. vesiculosus* group may be due to the lack of compliance. The amount of seaweed (extract) used in the literature to induce the hypocholesterolemia effects is relatively high as compared to the amount that was used in this study. Our results are in line with those of a recent placebo-controlled, double-blind intervention study showing no effects of consumption of 4.8 g of spirulina and wakame for 17 days on lipid parameters in non-hypercholesteremic adult men and women [34]. A previous clinical trial already showed that the administration of higher daily dosages of seaweed, a total of 48 g per day for 4 weeks, did reduce fasting blood glucose, 2 h postprandial blood glucose, and TG, and increase HDLc [13]. A recently performed randomized-controlled trial, where patients with metabolic dysfunction-associated fatty liver disease (MAFLD) were supplemented twice daily for 6 months with a combination of isolated sulphated polysaccharide fucoidan (875 mg) and xanthophyll (875 mg), resulted in a lower TC and TG ratio and an increased amount of anti-diabetic adipokines, adiponectin, and leptin as compared to the control group [53]. Intake of fucoidan and fucoxanthin further led to an attenuation in MAFLD-mediated insulin resistance. Approximately 131 kg of *S. fusiforme* would be required to consume 1750 mg fucoxanthin, which is not a realistic amount [54]. However, the fact that 48 g of seaweed per day was found to affect glucose metabolism is supportive of a strong synergetic effect of various nutraceuticals present in seaweed and seems feasible in clinical practice. It is, therefore, likely that higher dosages of the brown seaweeds *F. vesiculosus* or *S. fusiforme* can affect glucose metabolism and blood lipids and help in T2DM disease management.

We tried to verify adherence with the interventions by measuring seaweed-derived plant sterols in the blood. However, the seaweed-derived fucosterol and saringosterol in serum remained below the detection limit, but brassicasterol increased in the *S. fusiforme* group during the intervention but not in the *F. vesiculosus* group, in line with the potential non-adherence in the latter group. Brassicasterol is often detectable in *Phaeophyta*, brown seaweeds such as *Sargassum asperifolium* [55], but also in terrestrial plants, so it is not a perfect measure of adherence with seaweed interventions since it can also reflect vegetable intake.

This pilot study has several limitations. First, this study had a relatively low sample size. Due to several study withdrawals, less than 36 participants were included in the final analyses. This has a potential effect on the study power and external validation, although this study was of an explorative nature. Moreover, due to the low sample size, the randomization could not prevent a slightly unequal distribution of sex and medication use over the three groups. However, no significant difference between both sexes was found for blood lipids and glucose levels at baseline. Third, there may have been recall bias in checking nutritional intake of the participants. All the nutritional data were derived from questionnaires. In the nutritional diaries, under-reporting was noticed; however, this was the same for all study groups. Fourth, it was not possible to check for compliance in an objective way during the intervention because the values of seaweed-derived plant sterols were below the detection limit. Fifth, in this study, the blood glucose levels of the majority of the participants was already well-regulated due to polypharmacy. It is possible that the effects of seaweed on top of several antidiabetic and lipid-lowering drugs are less evident. Finally, the included group was quite heterogeneous in diabetes medication, which in turn increased the generalizability.

A strength of the current study is that we determined for the first time, in a randomized, controlled manner, that a small consumable amount of crude *S. fusiforme* and *F. vesiculosus* could affect glucose regulation or risk factors for CVD. Our study was pragmatic in design, allowing for us to examine the effects of ingesting fresh seaweed as part of a healthy diet in a real live setting. Moreover, to our knowledge, we were the first to use a blinded CGM in a human seaweed intervention study to determine glucose parameters, giving a sophisticated view on trends and effects. Also, there is little research to date on the effects of fresh *S. fusiforme* and *F. vesiculosus* intake in overweight individuals with T2DM.

## 5. Conclusions

Our data show that small consumable amounts of the seaweed species *S. fusiforme* or *F. vesiculosus* on top of regular treatment did not affect average blood glucose levels as determined by CGM. Possible explanations of this lack of effect could be the compliance with the intervention or a too-low dosage of seaweed. This pilot trial teaches us that it is difficult to persuade people to ingest enough of these brown seaweed species (especially of *F. vesiculosus*) when offered in their natural form as part of a healthy diet. Moreover, in people with T2DM and polypharmacy, a more potent intervention, such as an extract with higher amounts of the effective compounds, may be needed to yield effects. Future randomized, controlled studies are needed on parameters of metabolic health using other seaweed species in higher doses and focusing on compliance with the interventions to take a step toward implementing this sustainable nutritional strategy in the treatment of people with T2DM.

## Figures and Tables

**Figure 1 nutrients-16-01837-f001:**
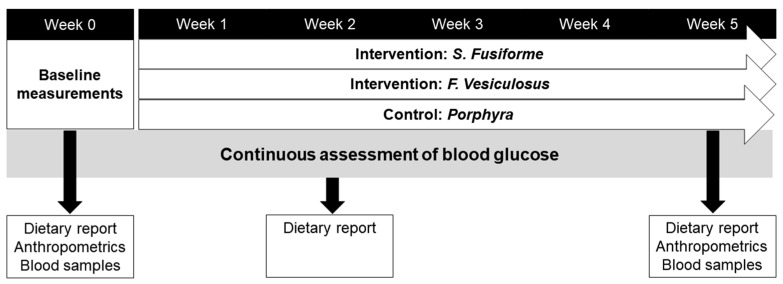
Timeline of study procedures.

**Figure 2 nutrients-16-01837-f002:**
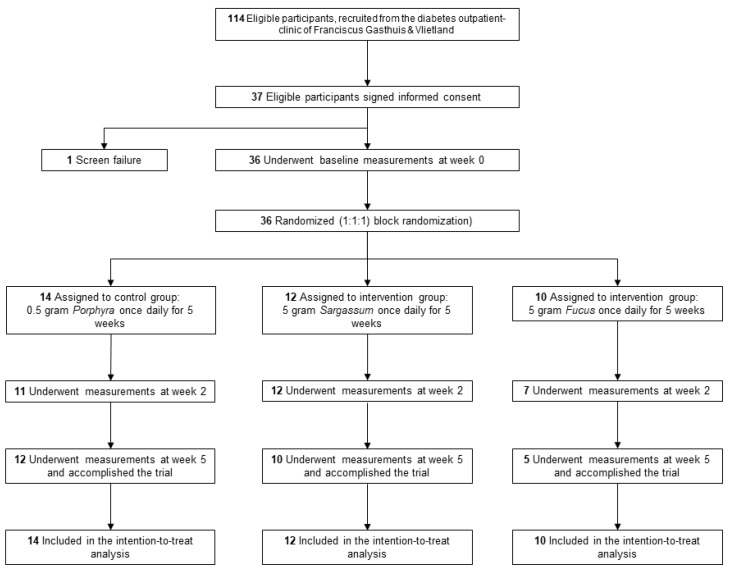
Study flowchart.

**Figure 3 nutrients-16-01837-f003:**
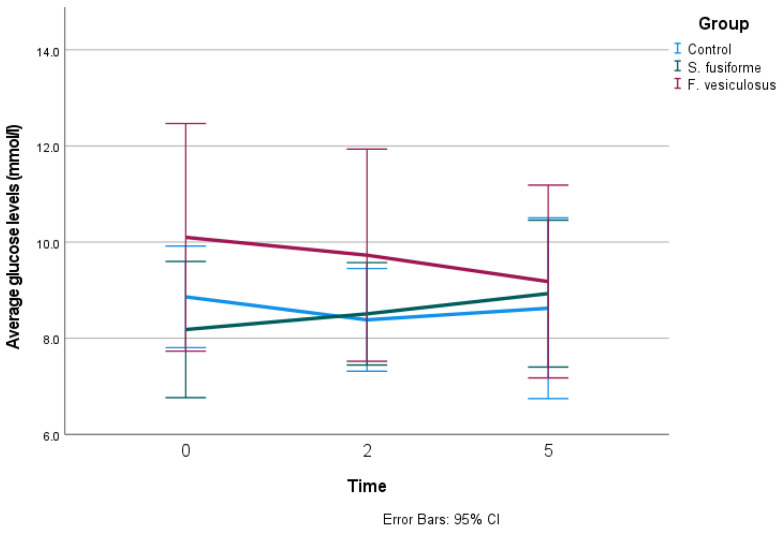
Effect of seaweed intake on weekly average glucose levels.

**Table 1 nutrients-16-01837-t001:** Chemical composition of the used dried *S. fusiforme* (Batch SFM201806), *F. vesiculosus* (Batch FVH201809) and *Porphyra*.

Chemical Composition	*S. fusiforme*	*F. vesiculosus*	*Porphyra*
Water (%)	15.0	9.29	6.5
Lipid (%)	4.23	6.43	1.5
Carbohydrate (%)	2.53	5.57	10.5
Arsenic (As) (%)	16.1	23.42	13.6
Mercury (Hg) (mg/kg)	0.055	0.016	-
Cadmium (Cd) (mg/kg)	0.361	0.282	-
Lead (Pb) (mg/kg)	0.347	0.512	-
Beryllium (Be) (mg/kg)	0.013	0.014	-
As (V, mg/kg)	16.86	36.8	-
iAs (V and III, mg/kg)	0.45	0.21	-
Phytosterols (%)	0.416	0.35	-

**Table 2 nutrients-16-01837-t002:** Baseline characteristics.

Parameter	Control (n = 14)	*S. fusiforme* (n = 12)	*F. vesiculosus* (n = 10)	*p*-Value
Sex, n (%) men	8 (57)	11 (92)	6 (60)	0.1
Age (Years)	65 (54–71)	60 (46–67)	66 (54–74)	0.7
Diabetes duration (years)	13 (11–21)	14 (11–19)	9 (6–14)	0.2
Insulin users, n (%)	10 (71)	8 (67)	4 (40)	0.3
Insulin use (units/day)	76 (57–108)	59 (46–98)	56 (15–74)	0.1
Antidiabetic drug use, n (%)	7 (50)	8 (67)	5 (46)	0.2
Body mass (kg)	103 ± 16.9	95 ± 7.5	102 ± 18.1	0.8
Energy intake	1543 ± 374	1748 ± 318	1428 ± 595	0.2
BMI (kg/m^2^)	34 ± 5.1	30 ± 2.8	34 ± 7.1	0.2
Waist circumference (cm)	119 ± 13.3	104 ± 5.5	115 ± 12.4	0.008
SBP (mm/Hg)	131 (127–148)	134 (125–137)	135 (125–146)	1.0
DBP (mm/Hg)	78 ± 7.8	82 ± 6.8	76 ± 3.7	0.1

Body mass index (BMI), systolic blood pressure (SBP), diastolic blood pressure (DBP).

**Table 3 nutrients-16-01837-t003:** Effect of seaweed administration on continuous blood glucose measurements.

Outcome	Group	Baseline	Week 2	Week 5	*p*-Value Within-Group Effect	*p*-Value Between-Group Effect over Time Compared to Control
Avg glucose (mmol/L)	Control (n = 12)	8.9 ± 1.7	8.4 ± 0.5	8.6 ± 0.9	0.78	
	*S. fusiforme* (n = 10)	8.2 ± 2.1	8.5 ± 0.5	9.0 ± 0.7	0.18	0.25
	*F. vesiculosus* (n = 5)	10.1 ± 3.3	9.7 ± 0.9	9.2 ± 0.7	0.86	0.90
Postprandial glucose (mmol/L)	Control (n = 8)	9.1 ± 1.6	9.5 ± 1.1	8.5 ± 0.6	0.64	
	*S. fusiforme* (n = 9)	9.7 ± 2.0	9.7 ± 1.0	9.6 ± 0.9	0.92	0.97
	*F. vesiculosus* (n = 2)	9.6 ± 2.9	9.5 ± 1.4	11.6 ± 2.1	0.37	0.92
TIR (%)	Control (n = 12)	69 ± 24	74 ± 7.1	76 ± 7.6	0.38	
	*S. fusiforme* (n = 10)	67 ± 23	68 ± 6.4	69 ± 8.2	0.58	0.87
	*F. vesiculosus* (n = 5)	55 ± 31	53 ± 10.4	58 ± 10.8	0.78	0.75
TAR (%)	Control (n = 12)	29 ± 24	24 ± 6.7	23 ± 7.8	0.44	
	*S. fusiforme* (n = 10)	26 ± 23	28 ± 6.5	29 ± 8.3	0.47	0.45
	*F. vesiculosus* (n = 5)	42 ± 33	43 ± 11.5	37 ± 11.3	0.63	0.86
TUR (%)	Control (n = 12)	0.0 (0–1.3)	0.5 (0–1.5)	0.7 (0–3.0)	0.26	
	*S. fusiforme* (n = 10)	0.0 (0–6.3)	0.6 (0–3.9)	0 (0–2.0)	0.50	0.39
	*F. vesiculosus* (n = 5)	0.0 (0–13.3)	0 (0–3.5)	0 (0–14)	0.18	0.40
eHbA1c % (mmol/mol)	Control (n = 13)	7.2 ± 1.1	-	6.8 ± 0.3	0.25	
	*S. fusiforme* (n = 11)	6.7 ± 1.4	-	7.2 ± 0.4	0.13	0.05
	*F. vesiculosus* (n = 9)	8.1 ± 2.2	-	7.4 ± 0.5	0.78	0.82

Avg = average weekly glucose; TIR = time in range; TAR = time above range; TUR = time under range; eHbA1c = calculated hemoglobin A1c.

**Table 4 nutrients-16-01837-t004:** The effect of *S. fusiforme* and *F. vesiculosus* on plasma lipid markers.

Outcome	Group	Baseline	Week 5	Within-Group Effect (*p*-Value)	Between-Group Effect over Time Compared to Control (*p*-Value)
TC (mmol/L)	Control (n = 10)	4.1 ± 0.8	3.9 ± 0.3	<0.05	
	*S. fusiforme* (n = 9)	4.0 ± 1.2	3.8 ± 0.2	0.17	0.48
	*F. vesiculosus* (n = 5)	3.7 ± 0.6	3.8 ± 0.4	0.47	0.08
TG (mmol/L)	Control (n = 10)	2.0 ± 1.0	1.7 ± 0.3	<0.05	
	*S. fusiforme* (n = 9)	2.7 ± 1.9	3.0 ± 0.8	<0.001	0.25
	*F. vesiculosus* (n = 5)	1.0 ± 0.4	1.1 ± 0.2	<0.05	0.06
HDLc (mmol/L)	Control (n = 10)	1.4 (0.9–1.5)	1.4 (0.8–1.6)	0.68	
	*S. fusiforme* (n = 9)	1.0 (0.8–1.4)	1.2 (0.7–1.5)	0.91	0.51
	*F. vesiculosus* (n = 5)	1.2 (1.2–1.6)	1.2 (1.2–1.5)	<0.05	0.93
LDLc (mmol/L)	Control (n = 10)	2.2 (1.9–2.6)	2.1 (1.7– 2.7)	0.68	
	*S. fusiforme* (n = 9)	1.9 (1.6–2.3)	1.9 (1.5–2.4)	0.95	0.93
	*F. vesiculosus* (n = 5)	2.2 (1.8–2.8)	2.1 (1.8–2.9)	0.89	0.89
Lp(a) (g/L)	Control (n = 10)	0.1 (0.07–0.22)	0.1 (0.09–0.30)	0.41	
	*S. fusiforme* (n = 10)	0.1 (0.08–0.13)	0.1 (0.07–0.21)	0.55	0.72
	*F. vesiculosus* (n = 5)	0.4 (0.21–0.49)	0.3 (0.1–0.5)	0.34	0.40
ApoA-I (g/L)	Control (n = 10)	1.6 (1.2–1.8)	1.5 (1.2–1.8)	<0.001	
	*S. fusiforme* (n = 10)	1.3 (1.2–1.4)	1.3 (1.2–1.4)	<0.05	0.78
	*F. vesiculosus* (n = 5)	1.2 (1.2–1.3)	1.3 (1.2–1.6)	<0.05	0.44
ApoB (g/L)	Control (n = 10)	0.8 ± 0.2	0.8 ± 0.08	<0.001	
	*S. fusiforme* (n = 10)	0.7 ± 0.2	0.7 ± 0.09	0.28	0.57
	*F. vesiculosus* (n = 5)	0.7 ± 0.2	0.6 ± 0.09	<0.05	0.89

TC = total cholesterol; TG = triglycerides; HDLc = high-density lipoprotein c; LDLc = low-density lipoprotein c; Lp(a) = lipoprotein a; ApoA-I = apolipoprotein A1; ApoB = apolipoprotein B.

**Table 5 nutrients-16-01837-t005:** The effect of brown seaweed on plasma sterol composition.

Outcome	Control		*S. fusiforme*		*F. vesiculosus*		Between-Group Effects (*p*-Value)	
	Baseline (n = 10)	Week 5 (n = 10)	Baseline (n = 10)	Week 5 (n = 10)	Baseline (n = 5)	Week 5 (n = 5)	Control vs. *S. fusiforme*	Control vs. *F. vesiculosus*
Lathosterol (mg/dL)	0.12 (0.01–0.23)	0.10 (0.07–0.12)	0.13 (0.06–0.19)	0.11 (0.05–0.17)	0.10 (0.05–0.11)	0.11 (0.07–0.15)	0.325	0.270
Campesterol (mg/dL)	0.46 ± 0.07	0.36 ± 0.06	0.48 ± 0.06	0.52 ± 0.07	0.60 ± 0.09	0.56 ± 0.08	0.017	0.228
Stigmasterol (µg/dL)	8.2 ± 1.7	7.1 ± 1.4	5.8 ± 0.8	6.5 ± 1.0	10 ± 2.1	9.9 ± 2.1	0.056	0.298
Sitosterol (mg/dL)	0.27 ± 0.04	0.22 ± 0.03	0.30 ± 0.03	0.30 ± 0.04	0.36 ± 0.06	0.35 ± 0.06	0.097	0.174
Avenasterol (µg/dL)	44 ± 5.1	38 ± 3.7	50 ± 5.5	50 ± 6.5	59 ± 10	61 ± 12	0.145	0.144
Brassicasterol (µg/dL)	10 (7.5–13)	6.4 (3.9–8.9)	8.6 (5.2–12)	12 (9.7–14)	11 (6.3–15)	12 (8.1–15)	0.003	0.192
Desmosterol (mg/dL)	0.16 (0.06–0.27)	0.12 (0.06–0.17) **	0.18 (0.05–0.30)	0.16 (0.07–0.25) **	0.11 (0.07–0.16)	0.13 (0.11–0.16)	0.097	0.050
24OH Cholesterol (ng/mL)	44 ± 4.2	39 ± 4.1	38 ± 4.5	37 ± 4.5	41 ± 1.0	39 ± 2.9	0.304	0.595
7αOH-Cholesterol (ng/mL)	105 ± 19	81.8 ± 11	125 ± 22	116 ± 20	93.3 ± 13	86.4 ± 12	0.391	0.351
27OH-Cholesterol (ng/mL)	128 ± 13	117 ± 13 **	114 ± 10	110 ± 13 **	133 ± 5.7	125 ± 11	0.461	0.786
Cholesterol (GC) (mg/dL)	185 ± 13	165 ± 14 **	171 ± 12	159 ± 9.3 **	174 ± 5.9	173 ± 20	0.460	0.394

Data are mean ± SEM or median (IQR); effects are relative to the control group. ** = within-group effect. α = 0.008 for Lathosterol–Desmosterol. α = 0.0125 for 24OH Cholesterol–Cholesterol (GC).

## Data Availability

The data presented in this study are available on request from the corresponding author due to privacy.

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
