# Peer review of "The Effect of Sargassum fusiforme and Fucus vesiculosus on Continuous Glucose Levels in Overweight Patients with Type 2 Diabetes Mellitus: A Feasibility Randomized, Double-Blind, Placebo-Controlled Trial"

_nutrients, 2024, doi:10.3390/nu16121837_

Round 1
Reviewer 1 Report
Comments and Suggestions for Authors
Outcome data for group F. vesiculosus are missing about 50% partecipants (Table 3). Please explain.
In table 3 the authors reported the effect of seaweed administration on continous blood glucose measurements at Baseline, Week 2 and Week 5.
In Figure 3 the authors reported the Effect of seaweed intake on weekly average glucose levels at Baseline, Week 3 and Week 5.
Please present the same follow-up (week 2 or week 3, or all weeks in figure and only baseline and week 5, as for the other outcomes presented in Table 4)
Author Response
Thank you for the feedback to improve our manuscript.
- Outcome data for group F. vesiculosus are missing about 50% participants (Table 3). Please explain.
In the results section line 230-235 we explain why these data are missing:
‘Reasons for withdrawal were: 1) gastrointestinal complaints (n=3; F. vesiculosus), 2) the study being too demanding (n=4; 2 control, 1 F. vesiculosus, 1 S. fusiforme), 3) allergic reaction to band-aid used for the CGM (n=1; F. vesiculosus) and, 4) terminal illness of a family member (n=1; S. fusiforme).’
Also, in the discussion we comment on the drop-outs in the F. vesiculosus group (lines 377-381):
‘Three people in the F. vesiculosus group stopped the intervention because of gastrointestinal complaints, while in the S. fusiforme and control group, there were no withdrawals related to gastrointestinal complaints. Therefore, it is likely that F. vesiculosus is not as well tolerated and not as well appreciated in its natural form, so that the actual dose taken by the participants during the study was much lower than instructed and did not result in health effects.’
- In table 3 the authors reported the effect of seaweed administration on continuous blood glucose measurements at Baseline, Week 2 and Week 5. In Figure 3 the authors reported the Effect of seaweed intake on weekly average glucose levels at Baseline, Week 3 and Week 5. Please present the same follow-up (week 2 or week 3, or all weeks in figure and only baseline and week 5, as for the other outcomes presented in Table 4).
We thank the reviewer for pointing out this error, we now changed the week number to week 2 as reported in the flow diagram.
Reviewer 2 Report
Comments and Suggestions for Authors
The paper “The effect of Sargassum fusiforme and Fucus vesiculosus on continuous glucose levels in overweight patients with type 2 diabetes mellitus: a feasibility randomized, double-blind, placebo-controlled trial” will to the growth of the literature on research, especially for nutritionists, doctors and food producers, especially nutrient-enriched food for diabetes.
However, the following items should be revised:
Introductions
The authors did not describe.
Are there groups at risk of the effects of high seaweed consumption?
- whether there is no such risk?
- what about the iodine content?
- what about pregnant women?
Materials and Methods
Line 216
(p<0.05) - it should be Italic - p<0.05 - the same in the next ones
Results
Title of Fig. 3 and Tab. 3 - I suggest giving the names of the seaweed (as in the title).
The data in table 3
I suggest averaging results to the same levels (same accuracy) e.g. 0.9 0.97 for the same indicator, e.g. p-value
Similar to Tables 4 and 5.
The title of Table 5 is not clear
What products are they concerned with the content of the indicated ingredients?
“Sterol content, endogenous cholesterol markers, metabolites and precursors” in?
Discussion
“None of those 3 studies included S. fusiforme or S. vesiculosus but utilized the brown seaweed Undaria pinnatifida. In subgroup analyses, the use of fresh seaweed was related to improved post- prandial blood glucose levels, contrary to our findings.” - Was the composition of fresh and dried seaweed similar? Is there a danger that the drying process could have resulted in the inactivation of certain ingredients?
Were people's diets monitored and compared in the conclusions throughout the study, or were they the same all the time? Or maybe people consuming Porphyra had more control over what they ate during the study?
Conclusion
"This pilot trial teaches us that it is difficult to get people to ingest enough of these brown seaweed species (especially of F. vesiculosus), when offered in its natural form, as part of a healthy diet.
and why and what were the reasons for the difficulty "to get people to ingest enough of these brown seaweed "
The authors do not describe in the results any problems.
Author Response
Thank you for your extensive review, we could improve our manuscript substantially.
The paper “The effect of Sargassum fusiforme and Fucus vesiculosus on continuous glucose levels in overweight patients with type 2 diabetes mellitus: a feasibility randomized, double-blind, placebo-controlled trial” will to the growth of the literature on research, especially for nutritionists, doctors and food producers, especially nutrient-enriched food for diabetes. However, the following items should be revised:
Introductions
- The authors did not describe. Are there groups at risk of the effects of high seaweed consumption?
- whether there is no such risk?
- what about the iodine content?
- what about pregnant women?
Seaweed is commonly consumed in small amounts. Consumption of high amounts is not desirable because of the possible higher content of iodine or heavy metals (similar to fish) of the seaweed, depending on the species. People with thyroid problems for example should not eat high amounts of seaweed species with a high iodine content, such as Kelps. Also, pregnant women should be careful concerning seaweed amounts and species (again similar to fish consumption). Iodine and heavy metals can also be removed from the seaweed by pretreatment, although not completely. That is why we excluded pregnancy and people with thyroid diseases, as we note in the exclusion criteria in lines 96-99 of the revised version of the manuscript:
Exclusion criteria were: type 1 or monogenetic forms of diabetes, thyroid disease, pregnancy, and usage of corticosteroids, usage of blood anti-coagulants, and history of heart failure or myocardial infarction within last 3 months, transplantation, or an allergy to shellfish.
Materials and Methods
- Line 216: (p<0.05) - it should be Italic - p<0.05 - the same in the next ones
We changed all the p-values as suggested.
Results
- Title of Fig. 3 and Tab. 3 - I suggest giving the names of the seaweed (as in the title).
Thank you for this suggestion, we changed the names in the figure.
- The data in table 3, 4 and 5 - I suggest averaging results to the same levels (same accuracy) e.g. 0.9 0.97 for the same indicator, e.g. p-value.
We made adjustments, so the accuracy is the same in all the tables.
- The title of Table 5 is not clear
The title is changed and reflects now more the content of the table.
Discussion
- “None of those 3 studies included S. fusiforme or S. vesiculosus but utilized the brown seaweed Undaria pinnatifida. In subgroup analyses, the use of fresh seaweed was related to improved post- prandial blood glucose levels, contrary to our findings.” - Was the composition of fresh and dried seaweed similar? Is there a danger that the drying process could have resulted in the inactivation of certain ingredients?
We added to the methods (line: 112-115): ‘The composition with respect to bioactive compounds may differ between fresh and dried seaweed. However, many bioactive components with health-improving properties have been demonstrated to remain intact. Among compounds of interest, the LXR-activating sterols, remain intact after drying (Bogie et al., 2019 and Martens et al., 2023).
- Were people's diets monitored and compared in the conclusions throughout the study, or were they the same all the time? Or maybe people consuming Porphyra had more control over what they ate during the study?
Yes, as described in the methods, intake was monitored using a food diary at baseline, week 2 and week 5. The food diaries were mainly used to calculate energy intake, to see whether the intervention had an influence on energy intake. In the results we describe:
‘In the intervention groups there was a non-significant self-reported decrease in energy intake in comparison with the control group (1748 ± 318 to 1603 ± 292 kcal for S. fusiforme, and 1428 ± 595 to 1231 ± 484 kcal for F. vesiculosus). There were no significant between-group differences when intervention groups were compared to control (S. fusiforme p=0.4 and F. vesiculosus p=0.7). No decrease in body weight was seen in both intervention groups.’
We amended this line for more clarity.
Except for the addition of seaweed to the normal diet, we did not recommend any changes in the participants' diet. However, it cannot be ruled out that intake changed during the intervention period. However, it is expected that people mainly adjusted the quantity (due to potential less appetite when using seaweed) and not the composition of their diet, nor is it expected that this would be different for the control group, a s we found only non-significant changes over time.
Conclusion
- "This pilot trial teaches us that it is difficult to get people to ingest enough of these brown seaweed species (especially of F. vesiculosus), when offered in its natural form, as part of a healthy diet. - and why and what were the reasons for the difficulty "to get people to ingest enough of these brown seaweed ". The authors do not describe in the results any problems.
In the results line 230-235 is explained why data were missing, in the F. vesiculosus group: 3 of the 5 drop-outs had gastrointestinal complaints. Also, in the discussion in lines 377 to 381 we commented on the non-compliance in this group.
Reviewer 3 Report
Comments and Suggestions for Authors
The effect of Sargassum fusiforme and Fucus vesiculosus on continuous glucose levels in overweight patients with type 2 diabetes mellitus: a feasibility randomized, double-blind, placebo controlled trial Geurts et al. This study was conducted to determine if the consumption of two different types of red algae Sargassum fusiforme (SF) and Fucus vesiculosus (FV) will reduce the symptoms associated with obesity and type -2 diabetes. The premise of this study was based on a number of published intervention trials in population based studies and in preclinical models in which either red algae as whole or its constituents were noted to alleviate the symptoms associated with diabetes and risk of cardiovascular diseases. The intervention duration was 5 weeks (the details are provided in the manuscript) and the authors concluded that there were no significant effect of intervention on the key parameters with a few exceptions. All results are presented as average with standard error of the mean at 95% confidence for statistical significance. Generally, the manuscript is sound. This manuscript is presented as a preliminary study outlining a detailed description of ethical approval, methodologies, discussion, outlining the limitations as well as the strength of the study and conclusion. The reviewer believes that this study which concludes that some of health benefits of functional components of our food must be carefully evaluated. The inclusion of high dosages of extracted components as opposed to the whole food must be considered. The reviewer has the following comments for authors to consider. 1. Some components of the text are ambiguous and a modified version with added clarity will strengthen the manuscript. Example, line 109,110 “the material was discarded…..” This statement is unclear as to who discarded the materials. Was this part of the method the authors followed? 2. With respect to FV a statement is made that “the standard was followed”. It is unclear as to what were the standards and how the criteria were determined. 3. A table should be provided outlining the level of various purported beneficial phytochemicals in SF, FV as well as in the placebo. In the current manuscript that information is missing. 4. In one table the concentration of various metals are provided. Are these levels present in the materials that were given to the participants, a clarification will be useful. 5. Lines 434-435 it is stated that possibly participants were non-adherent. It is difficult to support this statement without having knowledge of the levels of various phytosterols present in the test materials. 6. One observation the reviewer made is the following: In a majority of the tables reporting the results, baseline as opposed to post intervention values, the standard error is substantially smaller than the standard error in baseline values. Is this biologically relevant? It suggests that in each group there were inter-individual differences. Is it possible that some individuals responded to the intervention adversely or favorably
Author Response
The reviewer has the following comments for authors to consider.
- Some components of the text are ambiguous and a modified version with added clarity will strengthen the manuscript. Example, line 109,110 “the material was discarded…..” This statement is unclear as to who discarded the materials. Was this part of the method the authors followed?
Indeed, this was part of the protocol we followed, as earlier published by Yamashita et.al. We pretreated the seaweed according to the protocol of Yamashita to remove most of the inorganic Arsenic. We now made that clearer by changing the sentence in the methods (lines 108-112)
- With respect to FV a statement is made that “the standard was followed”. It is unclear as to what were the standards and how the criteria were determined.
We adjusted the sentence to make it clearer (lines 118-122)
- A table should be provided outlining the level of various purported beneficial phytochemicals in SF, FV as well as in the placebo. In the current manuscript that information is missing.
We added information on the phytosterol content of the two intervention Seaweeds used in Table 1.
- In one table the concentration of various metals are provided. Are these levels present in the materials that were given to the participants, a clarification will be useful.
Yes, this table is the chemical composition of the products given to the participant as mentioned in the title of the table.
We now also mention it in the text to make this more clear (lines 127-128).
- Lines 434-435 it is stated that possibly participants were non-adherent. It is difficult to support this statement without having knowledge of the levels of various phytosterols present in the test materials.
Because of the limited amount of the seaweed consumed by the participants and the relatively low amount of specific seaweed-derived phytosterols, the circulating concentration was below the detection limit and therefore, we could not draw any conclusions on adherence to the intervention. Moreover, the amount of phytosterols ingested cannot be translated directly into the levels found in the blood. On the other hand, the weekly visit of the participants to the hospital seemed to have improved adherence.
- One observation the reviewer made is the following: In a majority of the tables reporting the results, baseline as opposed to post intervention values, the standard error is substantially smaller than the standard error in baseline values. Is this biologically relevant? It suggests that in each group there were inter-individual differences. Is it possible that some individuals responded to the intervention adversely or favorably
Yes indeed, it could be pointing at inter individual differences, but our groups are too small to make a comment or conclusion on this matter. Future larger studies could be of help to explain this.
Reviewer 4 Report
Comments and Suggestions for Authors
This study proved that brown seaweed including Sargassum fusiforme and Fucus vesiculosus cannot relieve type 2 diabetes mellitus through a feasibility randomized, double-blind, placebo-controlled trial. Although brown seaweed does not alleviate the symptoms of diabetes patients, this provides a theoretical basis for the dietary intervention of brown seaweed to a certain extent. I have the following questions.
1. Whether the patients in control group were given 0.5 g Porphyra yezoensis? while the patients in S. fusiforme and F. vesiculosus groups were given 5 g S. fusiforme and F. vesiculosus, respectively.
2. In F. vesiculosus treatment group, three out of ten patients withdrew from the study due to gastrointestinal complaints. This probability is high. Is it possible that F. vesiculosus may cause damage to the intestines?
3. From the perspective of gender ratio, the grouping of patients seems unreasonable because the proportion of men in group F. vesiculosus reached to 92%, while in the other two groups, the proportion of men is 57% and 60%, respectively.
4. There are significant differences among the participants in this study, for example, some receive insulin and antidiabetic drugs for treatment, while others do not, this greatly affects the experimental results.
5. The author concluded that the main reason why brown seaweed including Sargassum fusiforme and Fucus vesiculosus do not have a hypoglycemic effect in this study is due to their low dosage. What is the maximum dose of brown seaweed that adults can consume daily, and what dose may be effective?
6. Sargassum fusiforme and Fucus vesiculosus have As, Hg, Cd, Pb. Is this normal?
7. In lines 64 and 65, the style of writing of α-amylase and α-glucosidase is inconsistent.
8. In line 120, it is 0.5 grams rather than 0,5 grams.
Author Response
Thank you for the comments to improve our manuscript.
This study proved that brown seaweed including Sargassum fusiforme and Fucus vesiculosus cannot relieve type 2 diabetes mellitus through a feasibility randomized, double-blind, placebo-controlled trial. Although brown seaweed does not alleviate the symptoms of diabetes patients, this provides a theoretical basis for the dietary intervention of brown seaweed to a certain extent. I have the following questions.
- Whether the patients in control group were given 0.5 g Porphyra yezoensis? while the patients in fusiformeand F. vesiculosus groups were given 5 g S. fusiforme and F. vesiculosus, respectively.
Indeed, we chose to include a control group that also received a type of seaweed, but only 1/10th the amount of the intervention groups. We did not expect effectiveness on the outcome measures from this small amount. We chose this control group because this way everyone was receiving seaweed, and the participants did not know if they were in the control group.
- In vesiculosus treatment group, three out of ten patients withdrew from the study due to gastrointestinal complaints. This probability is high. Is it possible that F. vesiculosusmay cause damage to the intestines?
There is no evidence or articles reporting intestinal problems resulting from F. vesiculosus, so we could not draw any conclusions on this matter. Though F. vesiculosus is edible, it is not always very much appreciated because it is typically tough and hard.
- From the perspective of gender ratio, the grouping of patients seems unreasonable because the proportion of men in group vesiculosusreached to 92%, while in the other two groups, the proportion of men is 57% and 60%, respectively.
It is true this group is not well balanced. In total there were more men than women included in our study, and because of the small number of participants the randomization could not prevent an slightly unbalanced allocation. We now made a comment on this in the discussion, line 450-452.
- There are significant differences among the participants in this study, for example, some receive insulin and antidiabetic drugs for treatment, while others do not, this greatly affects the experimental results.
The reviewer is right that the characteristics of the patients are heterogeneous with respect to medication use, but if there was an effect we wanted it to be generalizable to all patients with type 2 diabetes. Therefore, we did not set specified inclusion criteria for this matter. We now added a sentence commenting on this topic in the discussion section line 458-459.
- The author concluded that the main reason why brown seaweed including Sargassum fusiformeand Fucus vesiculosus do not have a hypoglycemic effect in this study is due to their low dosage. What is the maximum dose of brown seaweed that adults can consume daily, and what dose may be effective?
For S. fusiforme, 5 grams is considered a regular dose, as this amount is eaten on average per day in Japan. Since we show that this dose used on top of regular diabetes treatment is not sufficiently effective, for this group a high-dosed extract may be necessary to demonstrate an effect. We added this in the conclusion, lines 461-462.
- Sargassum fusiformeand Fucus vesiculosus have As, Hg, Cd, Pb. Is this normal.
Yes, this is common for seaweed. However, the content per species is highly variable and it can also be notably affected by the environment.
Pretreatment of S. fusiforme is very important as it removes most of the heavy metals, As in particular. Therefore, in the case of seaweed consumption it is highly important to consume quality-controlled batches.
- In lines 64 and 65, the style of writing of α-amylase and α-glucosidase is inconsistent and in line 120, it is 0.5 grams rather than 0,5 grams.
Thank you for noticing, we changed this in the revised manuscript.
Round 2
Reviewer 4 Report
Comments and Suggestions for Authors
The manuscript has been improved.
I still wonder why the doses of Porphyra yezoensis, S. fusiforme, and F. vesiculosus in the control group and the intervention group are different.
Author Response
Thank you for your additional question: 1. I still wonder why the doses of P. yezoensis, S. fusiforme, and F. vesiculosus in the control group and the intervention group are different. In our study, we studied 2 interventions: 5 mg S. fusiforme per day and 5 mg F. vesiculosus per day. We compared this with a control/placebo group. Since we wanted to conduct the study double-blind, the placebo also had to have the smell and taste of seaweed, but not the active ingredients. Therefore, we ended up using a very low dose of P. Yezoenis of 0.5 grams per day. This was to lower te chance of any bioactive compounds that might still have an effect in the control/placebogroup. We modified the manuscript a little more (line 107-109) to clarify this further.